# Effects of Medium- and Short-Chain Fatty Acids on Growth Performance, Nutrient Digestibility, Gut Microbiota and Immune Function in Weaned Piglets

**DOI:** 10.3390/ani15010037

**Published:** 2024-12-26

**Authors:** Shuang Dong, Nan Zhang, Jihua Wang, Yu Cao, Lee J. Johnston, Yongxi Ma

**Affiliations:** 1State Key Laboratory of Animal Nutrition, College of Animal Science and Technology, China Agricultural University, Beijing 100193, China; dongshuang19991106@163.com (S.D.); zhangnan1426@163.com (N.Z.); 2Galido Biotechnology Co., Ltd., Wuhan 430074, China; jeffrey_wang@calid.com.cn (J.W.); darcycao@foxmail.com (Y.C.); 3Department of Animal Science, West Central Research and Outreach Center, University of Minnesota, Morris, MN 56267, USA; johnstlj@morris.umn.edu

**Keywords:** α-glycerol monolaurate, glyceryl tributyrate, weaned piglets, growth performance, nutrient digestibility, immunological capacity, antioxidant capacity

## Abstract

α-Glycerol monolaurate (GML) and glyceryl tributyrate (TB) are representatives of medium- and short-chain fatty acids (MSCFA). GML hydrolyzed in the intestinal tract releases lauric acid, promoting fat digestion and absorption. TB, as a precursor of butyric acid, provides the main energy for intestinal epithelial cells, promotes their proliferation and repair, and strengthens the intestinal barrier function. Although these two fatty acids have shown promising results in animal production, their high additive levels and costs limit applications. In contrast, a mixture of low-dose GML and TB may be a potential novel nutritional modulation strategy. However, it is unclear whether the mixing of the two has synergistic effects. This study aimed to evaluate the effects of a mixture of GML and TB on the growth, serum indices, intestinal function, and gut microbial community of weaned piglets. The results indicated that MSCFA supplementation did not significantly improve the performance of weaned piglets. However, it can improve nutrient digestibility, enhance immunity and antioxidant capacity, improve intestinal health of piglets. These findings suggest that mixture of GML and TB is a nutrition regulation strategy worthy of further exploration in modern animal husbandry.

## 1. Introduction

Early weaning poses challenges to piglets, leading to intestinal and immune dysfunction, which results in malabsorption, diarrhea, and reduced growth performance [1]. Short- or medium-chain fatty acids can alleviate weaning stress in piglets [2,3]. α-glycerol monolaurate (GML) and glyceryl tributyrate (TB) have attracted considerable attention in recent years as nutritional modifiers. The glyceryl tributyrate extracted from butyric acid is chemically stable and lacks a distinctive odor. It is metabolized by lipase in the small intestine and gradually releases butyric acid, thereby exerting its biological activity [4]. The addition of TB to piglet diets has been demonstrated to stimulate appetite in weaned piglets by regulating protein [5] and lipid metabolism [6]. Furthermore, it has been proven to modulate the expression of intestinal barrier function genes and the production of inflammatory cytokines, thereby enhancing the ability of piglets to resist bacterial infections and further protecting the integrity of the intestinal morphology [7,8]. Additionally, other studies have demonstrated that diets supplemented with TB have a protective effect on dyslipidemia in mice [9] and lipopolysaccharide-induced liver injury in rats [10]. Lauric acid is a medium-chain fatty acid containing 12 carbon atoms, and its esterification with monoglycerides of lauric acid has been shown to exhibit a broad spectrum of antibacterial, antiviral, and anti-inflammatory effects [11,12,13]. Additionally, GML can effectively improve the absorption of nutrients and lipid metabolism of piglets to meet the energy requirements of weaned piglets [14]. Li et al. reported that supplementation with 0.1% α-GML reduces diarrhea rate, improves intestinal morphology, antioxidant capacity, immune status, and ameliorates gut microbiota in weaned pigs [15]. Moreover, supplementation with GML also improves the balance of the intestinal microbiota in broilers [16] and improves the performance of broilers [17]. These studies reported on single-component applications for piglet production with higher effective doses and higher supplement costs. Studies have shown that both GML and TB have positive effects on piglets, but their modes of action are different [17,18,19]. The question of whether synergistic effects can be achieved when TB and GML are mixed has not been addressed. Similar studies have demonstrated that supplementation with organic acids and medium-chain fatty acids is typically more effective than supplementation with a single acid due to the synergistic effect [20,21]. Therefore, the objective of this study was to investigate the effects of two low-dose compounds (GML and TB) on the growth performance, gut microbiota, and immune function of weaned piglets, with the aim of developing a novel nutritional regulation strategy for piglet production.

## 2. Materials and Methods

The experimental design and procedures used in this study were approved by the Animal Care and Use Committee of the Institute of China Agricultural University (Aw52104202-1-3; Beijing, China). This experiment was conducted in the Fengning Pig Experimental Base (Chengde, China).

### 2.1. Animals and Experimental Designs

A total of 120 weaned piglets [Duroc × (Landrace × Yorkshire), 28 days] with an initial average body weight of 6.88 kg were randomly assigned into three treatment groups according to body weight and gender. There were five pens in each treatment group, each housing eight piglets (four barrows and four gilts) for the 28-day experiment. Dietary treatments included (1) CON: a corn–soybean basal diet; (2) 0.1%: a basal diet with 0.1% MSCFA (GML/TB = 1:1); (3) 0.2%: a basal diet with 0.2% MSCFA (GML/TB = 1:1). GML (purity > 90%) and TB (purity > 60%, the left 40% is SiO_2_) were provided by Galido Biotechnology Co., Ltd. (Wuhan, China).

The corn–soybean basal diets were formulated to meet recommended requirements described as National Research Council (NRC, 2012) and are shown in Table 1 [22].

### 2.2. Feeding and Management

All piglets were housed in a temperature-controlled nursery (temperature 26~28 °C; humidity 55~70%) and were allowed to access feed and water freely. The troughs were checked daily at 08:30 and 15:30, and feed was added as needed for ad libitum, and their daily feed intake, feces, and health state were observed and recorded.

### 2.3. Sample Collections

Feces from each pen were collected from days 12 to 14 and days 26 to 28 and immediately frozen at −20 °C. Approximately 400 g of feces was collected by the grab sample technique and dried at 65 °C for 72 h. Dried samples were allowed to regain moisture for 24 h to ensure the accuracy of subsequent moisture determination. Fecal samples were ground to pass through a 1 mm sieve for further analysis.

On the morning of days 15 and 28, five piglets were selected randomly (one piglet per pen) from each treatment after 12 h of fasting for the collection of 10 mL serum. Blood samples from the jugular vein were collected into heparinized vacuum tubes and were centrifuged at 3000× *g* for 10 min at 4 °C to obtain serum. Serum samples were stored immediately at −20 °C.

On day 28, five piglets from each treatment group close to the median body weight were selected for slaughter. A 2 cm sample of the mid-section of the duodenum, jejunum, and ileum were collected, after removing its contents and washed with saline. Then, the intestinal samples were stored in 4% paraformaldehyde for 24 h for morphological examination. Intestinal mucosa (duodenum, jejunum, ileum) was scraped gently into centrifuge tubes 1.5 mL by using a sterile glass slide (Taizhou Huien Medical Equipment Co., LTD., Hiangsu, China). Mucosa samples were stored at −80 °C for further analysis of digestive enzyme activity. Cecum contents were collected in sterile containers and immediately stored in liquid nitrogen for analysis of the microbiota community.

### 2.4. Analysis of Growth Performance

Piglets were weighed individually on days 0, 14, and 28 to calculate the average daily growth (ADG). Feed consumption in each pen was recorded daily to calculate the average daily feed intake (ADFI) and the feed conversion ratio (FCR, ADFI/ADG).

### 2.5. Chemical Analysis for Diet and Feces

The dry matter (DM), ether extract (EE), ash, and crude protein (CP) contents of diets and feces were analyzed using the method of [23]. Gross energy (GE) was determined by an automatic isoperibolic oxygen bomb calorimeter (Parr 1281, Automatic Energy Analyzer; Moline, IL, USA). Organic matter (OM) was calculated as OM = 100 − ash. The AIA was determined using the methods described by McCarthy et al. [24]. The apparent total tract digestibility (ATTD) of dietary nutrients was calculated using the following equation:$$\mathrm{ATTD}\;\mathrm{of}\;\mathrm{nutrients}\;(\%)=100-({\mathrm{AIA}}_{\mathrm{diet}}\times {\mathrm{Nutrient}}_{\mathrm{feces}})/({\mathrm{AIA}}_{\mathrm{feces}}\times {\mathrm{Nutrient}}_{\mathrm{diet}})\;\times \;100$$

### 2.6. Serum Physiological and Biochemical Properties

Enzyme-linked immunosorbent assay (ELISA) was performed to measure total antioxidant capacity (T-AOC), superoxide dismutase (SOD), blood urea nitrogen (BUN), malonaldehyde (MDA), glutathione peroxidase (GSH-Px), catalase (CAT), alanine aminotransferase (ALT), aspartate aminotransferase (AST), glucose (GLU), total protein (TP), total cholesterol (TC), albumin (ALB), diamine oxidase (DAO), immunoglobulin A (IgA), immunoglobulin G (IgG), and immunoglobulin M (IgM) following the manufacturers’ instructions (Nanjing Jiancheng Bioengineering Institute, Nanjing, China). Detailed detection methods are described by Zhang et al. [25].

### 2.7. Analysis of Intestinal Index

The detection methods of immunoglobulin A (IgA), immunoglobulin G (IgG), immunoglobulin M (IgM), total antioxidant capacity (T-AOC), superoxide dismutase (SOD), malondialdehyde (MDA), glutathione peroxidase (GSH-Px) and catalase (CAT) of intestinal tissues (duodenum, jejunum and ileum) were the same as those in serum. The concentrations of interleukin-10 (IL-10), interleukin-6 (IL-6), and tumor necrosis factor-α (TNF-α) in intestinal tissues were determined by the commercial ELISA kit (Nanjing Jiancheng Bioengineering Institute). All procedures were performed according to the manufacturer’s instructions.

### 2.8. Analysis of Intestinal Morphology

Fixed intestinal (duodenum, jejunum, and ileum) samples were cleared, dehydrated, and embedded in paraffin wax. Tissues were sectioned at 5 µm thickness and installed on glass slides, and stained with hematoxylin and eosin. Villus heights of at least 12 randomly orientated villi and their adjoining crypts were measured with a light microscope at 40× combined magnification using an image processing and an analysis system (version 1; Leica Imaging Systems Ltd., Cambridge, UK).

### 2.9. Intestinal Mucosal Digestive Enzyme Activity

The activities of digestive enzymes including amylase (AMS), trypsin, chymotrypsin and lipase in duodenum, jejunum and ileum mucosa were determined using a commercially ELISA according to the instructions (Nanjing Jiancheng Institute of Bioengineering, Nanjing, China). Enzyme activity was normalized by protein concentration (U/mg).

### 2.10. Gut Microbiota

Cecum contents of ~0.25 g were used to extract total bacterial DNA using a QIAamp DNA Stool Mini Kit (Qiagen, Hilden, Germany) according to the manufacturer’s instructions. The V3–V4 region of the bacterial 16S rRNA gene was amplified using universal primers 338F (5′-ACTCCTACGGGAGGCAGCAG-3′) and 806R (5′-GGACTACHVGGGTWTCTAAT-3′) with the following amplification program (95 °C for 3 min, followed by 27 cycles of 95 °C for 30 s, 55 °C for 30 s, and 72 °C for 45 s, and final extension at 72 °C for 10 min). MiSeq Illumina Sequencing was performed. Raw tags were obtained by merging paired-end reads using the FLASH software (v1.2.11, http://ccb.jhu.edu/software/FLASH/, accessed on 22 August 2023). Quantitative Insights Into Microbial Ecology and UPARSE software (ver. 11, http://drive5.com/uparse/, accessed on 22 August 2023) were used to cluster operational taxonomic units (OTUs) with a 97% identity. Taxonomic annotation was analyzed using the Ribosomal Database Project database (80% confidence) and the taxonomic composition of the bacterial community was then analyzed.

### 2.11. Statistical Analysis

The normality of residuals and equal variances was checked using the UNIVARIATE procedure of SAS 9.4 (SAS Inst. Inc., Cary, NC, USA). Pen was used as experimental unit for the growth performance and nutrient digestibility. Individual piglet as experimental unit for serum parameters, intestinal morphology, and intestinal inflammatory factors. Data were analyzed using the GLM procedure of SAS followed by Tukey’s test, and the results were presented as mean values ± SEM. Significant differences were considered at *p* < 0.05, whereas 0.05 ≤ *p* < 0.10 was considered as a tendency.

## 3. Results

### 3.1. Growth Performance

The growth performance of weaned piglets is shown in Table 2. No significant difference was observed in BW, ADG, ADFI, and FCR of weaned piglets among the different groups; however, MSCFA supplementation increased the BW and ADG of piglets compared with the CON group from d 0 to 28 (*p* > 0.05).

### 3.2. Apparent Total Tract Digestibility of Nutrients

As shown in Table 3, piglets fed 0.1% MSCFA had higher (*p* < 0.05) ATTD of CP and GE on day 14 and ATTD of GE on day 28 compared with the CON group. However, supplementation with 0.2% MSCFA decreased (*p* < 0.05) the ATTD of DM, OM, and CP on day 14, while it increased (*p* < 0.05) the ATTD of EE on day 28 compared with the CON group.

### 3.3. Serum Biochemistry and Status Oxidant and Antioxidant

Dietary 0.2% MSCFA supplementation increased contents of GSH-Px, MDA and DAO on d 14 and concentration of SOD on d 28 in the serum compared with other two treatments (*p* < 0.05; Table 4). Additionally, the content of MDA in the 0.1% MSCFA group was significantly lower (*p* < 0.05; Table 4) compared to the 0.2% MSCFA group on d 28.

### 3.4. Serum Immune Function and Inflammatory Factors

On day 14, higher concentrations of IgG, IgM, IgA, TNF-α, and IL-10 were observed in piglets supplemented with 0.2% MSCFA compared to the CON group (*p* < 0.05; Table 5). The concentration of IL-6 was lower and the concentration of IgM was higher in piglets fed 0.1% MSCFA (*p* < 0.05; Table 5) compared with the CON group. On day 28, dietary supplementation with 0.2% MSCFA increased the levels of IgA and IL-10 (*p* < 0.05; Table 5), while reducing the concentrations of TNF-α, IL-1β, and IL-6 (*p* < 0.05; Table 5) compared with the CON group. Additionally, supplementation with 0.2% MSCFA decreased the concentrations of IL-1β and IL-6 compared with the CON group (*p* < 0.05; Table 5).

### 3.5. Intestinal Status Antioxidant

In the duodenum, supplementation with 0.2% MSCFA increased the levels of CAT, MDA, and SOD compared with the CON group (*p* < 0.05; Table 6). In the jejunum, supplementation with 0.1% MSCFA decreased the contents of CAT, GSH-Px, and MDA, supplementation with 0.2% MSCFA increased the contents of SOD and T-AOC compared with the CON group (*p* < 0.05; Table 6). In the Ileum, supplementation with 0.2% MSCFA increased the SOD concentration while decreasing the CAT and T-AOC concentrations compared with the CON group (*p* < 0.05; Table 6).

### 3.6. Digestive Enzymes

Dietary supplementation with MSCFA increased the activity of lipase in duodenum (*p* < 0.05; Table 7) compared with the CON group. However, 0.1% MSCFA supplementation reduced lipase activity in the ileum compared with the CON group (*p* < 0.05; Table 7).

### 3.7. Intestinal Morphology

As shown in Table 8, dietary MSCFA supplementation had no significant effect on villus height, crypt depth and the ratio of villus height to crypt depth in the duodenum, jejunum and ileum of weaned piglets.

### 3.8. Gut Microbiota Community

Firmicutes and Bacteroidota were the dominant phyla in the fecal microbiota, accounting for approximately 90%, followed by Actinobacteria and Spirochaetota (Figure 1B). The relative abundance of Actinobacterota increased in piglets fed MSCFA compared with the CON group (Figure 1D). At the family level, *Lactobacillaceae* and *Prevotellaceae* were the most abundant, followed by *Lachnospiraceae*, *Ruminococcaceae*, *Clostridiaceae*, *Acidaminococcaceae*, and *Oscillospiraceae* (Figure 1A). The relative abundance of *Clostridiaceae* increased in piglets fed MSCFA compared with the CON group (*p* < 0.05; Figure 1C).

## 4. Discussion

The addition of medium- and short-chain fatty acids to the diet can not only provide an energy source for piglets but also improve their digestive function and enhance their immune function [15,26]. However, previous studies focused on the effects of a single addition of a certain fatty acid on the growth performance and health of piglets, and there were few studies on the impact of mixed addition. This study aimed to investigate the effects of mixed addition of TB and GML on the health of piglets. Our results showed that 0.1% MSCFA in the diet can increase the ADG of weaned piglets. Therefore, adding MSCFA can improve the growth of weaned piglets to a certain extent.

Dietary supplementation with GML and TB improved growth performance by regulating nutrients and lipid metabolism [5,6,14]. In the present study, the results showed that dietary supplementation with 0.1% MSCFA improved the growth performance of piglets to a certain extent compared with the CON group, which may be attributed to the improvement of nutrient digestibility. Our results also showed that supplementing 0.1% MSCFA significantly increased the ATTD of CP and GE on day 14, and the ATTD of GE on day 28 compared with the CON group. Dierick et al. [27] showed that medium-chain fatty acids significantly increased small intestinal villus height and reduced crypt depth, thereby enhancing nutrient absorption in weaned piglets. Similarly, Cera et al. [28] found that adding coconut oil (rich in GML) increased lipase activity in the intestine, improving fat digestibility. This is consistent with our findings that MSCFA supplementation improves crude fat digestibility in the later period of this study. However, supplementation with 0.2% MSCFA had no significant effect on the growth performance and nutrient digestibility of piglets, which may be due to excessive addition. Supplementing 0.5% TB improved the growth [29], but 1.0% TB negatively affected the growth of weaned piglets [30]. Moreover, Snoeck et al. [31] and Fortuoso et al. [32] found that GML improved body weight in weaned piglets, while increasing the GML concentration did not further enhance growth performance [33,34], which was in agreement with our results.

The antioxidant defense system in piglets consists mainly of antioxidant enzymes and bio antioxidants, including T-AOC and SOD [35]. Malondialdehyde (MDA), produced through lipid peroxidation, serves as an indicator of lipid oxidation [36]. The present study showed that supplementing with 0.1% MSCFA increased T-AOC and SOD levels in the duodenum, jejunum, and ileum, while decreasing MDA levels in the jejunum and serum, indicating MSCFA supplementation might have boosted the oxidative stress defense system. Previous studies have shown that dietary GML reduces MDA concentrations, thus playing a key role in lowering lipid peroxidation and enhancing antioxidant capacity [15]. Kong et al. [37] found that GML decreased MDA content in serum and jejunum, and increased T-SOD and T-AOC activities by reducing inflammation and modulating the TLR4/NF-κB pathway. Furthermore, Wang et al. [38] showed that dietary tributyrin reduces MDA and H_2_O_2_ levels, alleviating intestinal oxidative stress in weaned piglets. Our results showed a synergistic antioxidant effect of TB and GML, while higher doses of MSCFA did not show better effects.

During the early postweaning period, antibody-mediated immune responses are crucial for the health and growth of piglets and the ability of piglets to acquire IgG through endocytosis ceases 24–36 h after birth [39]. Studies have shown that GML can influence immune cell function and regulate immunoglobulin production through interactions with membrane receptors [40], and Wu et al. found that short-chain fatty acids can promote IgA class conversion and production in intestinal B cells mediated by GPR43 [41].In this study, supplementation of MSCFA increased levels of IgA, IgM, and IgG in serum, indicating that MSCFA supplementation boosts immunoglobulin production, which was consistent with previous studies [42,43]. Furthermore, supplementation with MSCFA increased concentration of the anti-inflammatory cytokine IL-10 and decreased the pro-inflammatory cytokines such as TNF-α, IL-1β, and IL-6. Therefore, our results demonstrated that supplementation with MSCFA can improve the immunity status of piglets.

The activity of digestive enzymes is related to the digestion of nutrients. Our study found that 0.1% MSCFA supplementation significantly increased lipase activity in the duodenum and trypsin activity in the jejunum and ileum of piglets. These findings are supported by Kasprowicz-Potocka et al. [44]. Dietary GML or TB can be broken down to lauric and butyric acids by microbial action, improving intestinal pH [14,45]. In an acidic gut environment, digestive enzymes exhibit higher activity [46]. Furthermore, medium- and short-chain fatty acids, along with related oils, are absorbed directly by intestinal epithelial cells, enhancing intestinal morphology by increasing villus height, stimulating enzyme secretion, and improving membrane-bound enzyme activity [27,47,48,49]. In summary, MSCFA supplementation enhances digestive enzyme activity, likely due to the creation of a favorable pH environment in the gut.

Intestinal villus height, crypt depth, and their ratio are key indicators of intestinal health and function in piglets. Higher villus height and lower crypt depth enhance digestive enzyme activity and nutrient absorption [50,51]. Dietary GML increases lactic acid bacteria, inhibits harmful bacteria by regulating pH, and improves intestinal morphology in piglets [14,52]. Additionally, GML supplementation upregulates the expression of claudin-1, occluding, and ZO-1 proteins in the jejunum and ileum, thus protecting the intestinal barrier [34]. TB is metabolized to butyric acid by bacteria, which promotes the growth, proliferation, and differentiation of intestinal mucosal cells, enhancing intestinal barrier function [45]. In this study, MSCFA supplementation did not significantly improve villus height or the ratio of villus height to crypt depth. The discrepancies between our results and the above studies may be attributable to differences in the levels of MSCFA or the health status of the piglets.

In this study, Firmicutes and Bacteroidetes accounted for around 85% of the relative abundance. MSCFA supplementation increased the abundance of Firmicutes and decreased that of Bacteroidetes at the family level. This is consistent with previous research on the microbiota of the pig gastrointestinal tract [42,53]. Lactobacillus, a beneficial microorganism in Firmicutes phylum, plays a crucial role in regulating intestinal health and promoting growth by inhibiting harmful microbes [54]. Our results showed that Lactobacillus abundance increased in piglets fed MSCFA. Thus, MSCFA supplementation alters gut microbiota composition and supports the colonization of beneficial bacteria in weaned piglets.

## 5. Conclusions

In conclusion, dietary supplementation of MSCFA can improve nutrient digestibility, enhance immunity and antioxidant capacity, improve intestinal health and promote the colonization of beneficial bacteria in weaned piglets. The combined use of the two can give full play to their synergistic effect and improve the overall health status and production performance of piglets, which is a nutrition regulation strategy worthy of further exploration in modern animal husbandry. Therefore, future research should focus on the interrelationships and combined effects of more medium and short chain fatty acids, with particular emphasis on optimizing their combined effects at different intervention doses and timing to further enhance piglet growth performance and health.

## Figures and Tables

**Figure 1 animals-15-00037-f001:**
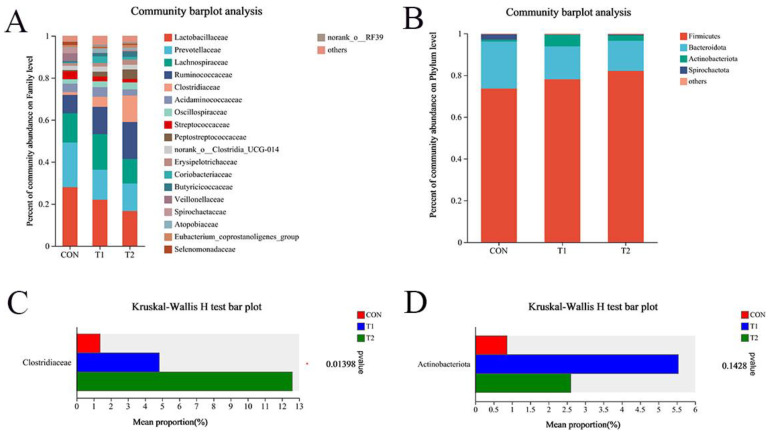
Differences in fecal microbial family level and phylum level of weaned piglets. (**A**,**C**), Fecal microbial composition of weaned piglets at family level; (**B**,**D**), Fecal microorganisms of weaned piglets differ at the phylum level. CON = a basal diet; 0.1% = a basal diet with 0.1% MSCFA (GML/TB = 1:1); 0.2% = a basal diet with the 0.2% MSCFA (GML/TB = 1:1); Data were means ± SD (*n* = 5); * *p* < 0.05 compared with the CON group.

**Table 1 animals-15-00037-t001:** Composition and nutrient levels of basal diets (as-fed basis, %).

Items	Content (%)
Ingredients	
Corn	63.48
Soybean meal, 43%	17.00
Extruded soybean	5.00
Soybean oil	1.50
Fish meal	5.00
Whey powder	4.00
CaHPO_4_	1.00
Limestone	0.70
L-lysine hydrochloride, 78%	0.50
L-threonine, 98%	0.20
DL-methionine, 98%	0.07
L-tryptophan, 98%	0.05
Choline chloride, 98%	0.30
Diatomite	0.20
NaCl	0.50
Premix ^1^	0.50
Total	100.00
Nutrient levels ^2^	
Metabolism energy, ME, MJ/kg	14.20
Crude protein, CP, %	18.51
Ether extract, EE, %	3.49
Dry matter, DM, %	89.83
Organic matter, OM, %	84.21
SID ^3^ Lysine, %	1.28
SID ^3^ Threonine, %	0.78
SID ^3^ Methionine, %	0.37
SID ^3^ Tryptophan, %	0.21

^1^ The premix provided the following per kilogram of diet: vitamin A, 12,000 IU; vitamin D_3_, 2000 IU; vitamin E, 30 IU; vitamin K_3_, 3 mg; vitamin B_1_, 3 mg; vitamin B_2_, 10 mg; vitamin B_6_, 6 mg; vitamin B_12_, 24 μg; nicotinic acid, 30 mg; D-pantothenic acid, 30 mg; folic acid, 2 mg; biotin, 0.3 mg; choline chloride, 600 mg; Fe, 120 mg; Cu, 10 mg; Mn, 35 mg; Zn, 120 mg; I, 0.3 mg; Se, 0.3 mg. ^2^ Metabolism was calculated according to NRC (2012). All others are measured values. ^3^ SID, standardized ileal digestible.

**Table 2 animals-15-00037-t002:** Effects of MSCFA on growth performance in weaned piglets.

Items	Treatments			SEM	*p*-Value
	CON	0.1%	0.2%		
Body wt 0 d, kg	6.68	6.68	6.67	1.18	1.00
Body wt 14 d, kg	10.18	10.96	10.74	1.43	0.74
Body wt 28 d, kg	16.02	17.34	16.48	2.32	0.71
day 1 to 14					
ADG g/d	285.60	304.73	290.45	28.40	0.60
ADFI g/d	474.52	467.56	458.04	56.13	0.91
FCR	1.58	1.52	1.58	0.08	0.60
day 14 to 28					
ADG g/d	413.76	437.14	418.13	74.72	0.89
ADFI g/d	638.61	744.27	666.16	113.31	0.36
FCR	1.70	1.71	1.59	0.09	0.81
Day 0 to 28					
ADG g/d	351.27	370.94	354.28	48.95	0.81
ADFI g/d	578.32	605.92	562.1	93.36	0.87
FCR	1.59	1.63	1.58	0.10	0.43

ADG = average daily gain; ADFI = average daily feed intake; FCR = feed conversion ratio; SEM = standard error of the mean (*n* = 5); CON = a basal diet; 0.1% = a basal diet with 0.1% MSCFA; 0.2% = a basal diet with the 0.2% MSCFA.

**Table 3 animals-15-00037-t003:** Effects of MSCFA on nutrient apparent digestibility in weaned piglets.

Items	Treatments			SEM	*p*-Value
	CON	0.1%	0.2%		
Day 14					
DM %	77.76 ^a^	80.17 ^a^	75.07 ^b^	2.63	<0.01
OM %	82.03 ^a^	84.36 ^a^	79.61 ^b^	2.41	<0.01
CP %	67.39 ^b^	72.68 ^a^	63.36 ^c^	4.49	<0.01
EE %	39.50	39.99	37.94	1.55	0.17
GE %	76.99 ^b^	79.92 ^a^	74.24 ^b^	2.88	<0.01
Day 28					
DM %	76.78	79.62	76.75	1.90	0.09
OM %	81.96	83.63	81.15	1.68	0.09
CP %	66.08	70.59	65.52	3.26	0.20
EE %	34.88 ^b^	37.17 ^ab^	39.42 ^a^	5.71	0.04
GE %	75.83 ^b^	79.15 ^a^	75.98 ^b^	2.10	0.05

DM = dry matter; OM = organic matter; CP = crude protein; EE = ether extract; GE = gross energy; SEM = standard error of the mean (*n* = 5); CON = a basal diet; 0.1% = a basal diet with 0.1% MSCFA (GML/TB = 1:1); 0.2% = a basal diet with the 0.2% MSCFA (GML/TB = 1:1). ^a,b,c^ Mean values within a row with different letters differ at *p* < 0.05. And 0.05 ≤ *p* < 0.10 was considered as a tendency.

**Table 4 animals-15-00037-t004:** Effects of MSCFA on serum biochemistry and status antioxidant in weaned piglets.

Items	Treatments			SEM	*p*-Value
	CON	0.1%	0.2%		
Day 14					
Biochemistry indexes					
TP g/L	37.12	40.65	42.58	6.69	0.60
AST U/L	43.44	32.43	50.41	16.22	0.27
ALT U/L	43.04	33.74	38.50	10.11	0.42
BUN mg/dL	5.35	6.05	4.33	1.68	0.28
ALB g/L	10.41	9.65	10.99	1.94	0.58
TC mmol/L	1.04	1.09	1.18	0.25	0.76
GLU mmol/L	3.74	4.35	4.59	0.66	0.08
Antioxidant indexes					
T-AOC U/mL	10.40	11.15	10.15	0.97	0.25
SOD U/mL	55.22	56.44	59.65	5.97	0.51
GSH-Px U/mL	314.24 ^b^	326.62 ^b^	371.13 ^a^	32.49	<0.01
MDA nmol/mL	3.35 ^b^	3.31 ^b^	4.74 ^a^	0.77	<0.01
CAT U/mL	39.62	46.22	41.69	8.63	0.5
DAO U/mL	1.66 ^b^	1.65 ^b^	2.44 ^a^	0.51	<0.01
Day 28					
Biochemistry indexes					
TP g/L	56.89	52.04	49.64	8.66	0.31
AST U/L	50.76	40.55	38.68	10.11	0.25
ALT U/L	41.71	38.90	34.33	9.08	0.50
BUN mg/dL	6.61	4.78	6.57	1.91	0.25
ALB g/L	11.43	10.62	9.96	2.45	0.54
TC mmol/L	1.67	1.55	1.41	0.31	0.46
GLU mmol/L	5.16	4.87	4.46	0.88	0.48
Antioxidant indexes					
T-AOC U/mL	10.03	9.84	10.41	0.56	0.34
SOD U/mL	56.38 ^b^	56.93 ^b^	65.84 ^a^	6.94	0.03
GSH-Px U/mL	342.88	350.71	366.68	15.92	0.06
MDA nmol/mL	4.10 ^ab^	3.38 ^b^	4.64 ^a^	0.73	<0.01
CAT U/mL	33.23	40.23	32.12	8.06	0.23
DAO U/mL	3.38	1.70	3.90	1.67	0.08

AST = aspartate aminotransferase; ALT = alanine aminotransferase; BUN = blood urea nitrogen; TP = total protein; ALB = albumin; TC = total cholesterol; GLU = glucose; GSH-Px = glutathione peroxidase; SOD = superoxide dismutase; T-AOC = total antioxidant capacity; MDA = malonaldehyde; CAT = catalase; DAO = diamine oxidase; SEM = standard error of the mean (*n* = 5); CON = a basal diet; 0.1% = a basal diet with 0.1% MSCFA (GML/TB = 1:1); 0.2% = a basal diet with the 0.2% MSCFA (GML/TB = 1:1). ^a,b^ Mean values within a row with different letters differ at *p* < 0.05. And 0.05 ≤ *p* < 0.10 was considered as a tendency.

**Table 5 animals-15-00037-t005:** Effects of MSCFA on serum immune function and inflammatory factors in weaned piglets.

Items	Treatments			SEM	*p*-Value
	CON	0.1%	0.2%		
Day 14					
Immune indexes					
IgG g/L	17.18 ^b^	19.57 ^ab^	21.81 ^a^	2.50	0.01
IgM g/L	1.28 ^c^	2.11 ^b^	2.83 ^a^	0.72	<0.01
IgA g/L	0.85 ^b^	0.87 ^b^	1.37 ^a^	0.30	<0.01
Inflammatory indexes					
TNF-α pg/mL	39.67 ^b^	39.27 ^b^	50.73 ^a^	7.40	0.01
IL-1β pg/mL	38.40	37.11	32.45	5.66	0.22
IL-6 pg/mL	117.14 ^a^	104.83 ^b^	117.82 ^a^	9.03	0.02
IL-8 pg/mL	40.83	39.52	36.16	4.20	0.20
IL-10 pg/mL	11.05 ^b^	12.72 ^b^	18.25 ^a^	3.46	<0.01
Day 28					
Immune indexes					
IgG g/L	17.62	19.41	16.10	2.38	0.10
IgM g/L	2.14	2.76	2.85	0.59	0.11
IgA g/L	1.38 ^b^	1.17 ^b^	1.77 ^a^	0.30	<0.01
Inflammatory indexes					
TNF-α pg/mL	47.33 ^a^	46.47 ^a^	33.59 ^b^	7.72	<0.01
IL-1β pg/mL	37.31 ^a^	32.88 ^b^	21.68 ^c^	6.85	<0.01
IL-6 pg/mL	128.02 ^a^	112.33 ^b^	112.53 ^b^	9.64	<0.01
IL-8 pg/mL	36.39	25.74	32.34	6.94	0.10
IL-10 pg/mL	16.92 ^b^	18.34 ^ab^	22.30 ^a^	3.52	0.02

IgG = immunoglobulin G; IgM = immunoglobulin M; IgA = immunoglobulin A; TNF-α = tumor necrosis factor-α; IL-1β = interleukin-1 beta; IL-6 = interleukin-6; IL-8 = interleukin-8; IL-10 = interleukin-10; SEM = standard error of the mean (*n* = 5); CON = a basal diet; 0.1% = a basal diet with 0.1% MSCFA (GML/TB = 1:1); 0.2% = a basal diet with the 0.2% MSCFA (GML/TB = 1:1). ^a,b,c^ Mean values within a row with different letters differ at *p* < 0.05. And 0.05 ≤ *p* < 0.10 was considered as a tendency.

**Table 6 animals-15-00037-t006:** Effects of MSCFA on antioxidant status in intestinal of weaned piglets.

Items	Treatments			SEM	*p*-Value
	CON	0.1%	0.2%		
Duodenum					
CAT U/mL	16.43 ^b^	18.87 ^b^	27.20 ^a^	3.05	<0.01
GSH-Px U/mL	341.66	333.38	355.39	35.65	0.65
MDA nmol/ml	3.19 ^b^	3.12 ^b^	5.77 ^a^	1.34	<0.01
SOD U/mL	46.98 ^b^	47.38 ^b^	58.66 ^a^	6.98	<0.01
T-AOC U/mL	6.29 ^ab^	7.91 ^a^	4.64 ^b^	1.76	0.01
Jejunum					
CAT U/mL	35.88 ^a^	23.42 ^b^	34.27 ^ab^	6.50	0.01
GSH-Px U/mL	492.25 ^a^	334.09 ^b^	357.55 ^b^	79.17	<0.01
MDA nmol/ml	5.43 ^a^	1.77 ^c^	3.81 ^b^	1.54	<0.01
SOD U/mL	40.77 ^b^	43.59 ^b^	58.97 ^a^	9.14	<0.01
T-AOC U/mL	7.50 ^b^	8.02 ^b^	10.50 ^a^	0.89	<0.01
Ileum					
CAT U/mL	29.56 ^a^	23.90 ^ab^	18.67 ^b^	5.40	<0.01
GSH-Px U/mL	455.35	420.27	372.02	60.43	0.08
MDA nmol/ml	4.69	4.30	4.40	0.78	0.76
SOD U/mL	47.25 ^b^	51.76 ^ab^	53.53 ^a^	3.87	0.03
T-AOC U/mL	6.98 ^a^	7.11 ^a^	4.73 ^b^	1.44	<0.01

CAT = catalase; GSH-Px = glutathione peroxidase; MDA = malonaldehyde; T-AOC = total antioxidant capacity; SOD = superoxide dismutase; SEM = standard error of the mean (*n* = 5); CON = a basal diet; 0.1% = a basal diet with 0.1% MSCFA (GML/TB = 1:1); 0.2% = a basal diet with the 0.2% MSCFA (GML/TB = 1:1). ^a,b,c^ Mean values within a row with different letters differ at *p* < 0.05. And 0.05 ≤ *p* < 0.10 was considered as a tendency.

**Table 7 animals-15-00037-t007:** Effects of MSCFA on intestinal mucosal digestive enzymes in weaned piglets.

Items	Treatments			SEM	*p*-Value
	CON	0.1%	0.2%		
Duodenum					
AMS U/mg	1.97	1.73	2.95	1.01	0.12
Trypsin U/mg	2454.77	1855.26	2025.32	465.74	0.11
Lipase U/L	39.06 ^b^	87.16 ^a^	71.01 ^a^	26.81	<0.01
Chymotrypsin U/mg	5.55	3.83	6.20	1.77	0.12
Jejunum					
AMS U/mg	0.99	1.08	1.19	0.27	0.61
Trypsin U/mg	2367.88	3050.92	2912.91	679.38	0.28
Lipase U/L	61.61	72.84	88.95	18.83	0.15
Chymotrypsin U/mg	3.53	3.69	4.20	1.18	0.65
Ileum					
AMS U/mg	1.68	1.48	1.24	0.41	0.26
Trypsin U/mg	3135.60	3384.44	2907.17	433.67	0.32
Lipase U/L	88.91 ^a^	72.07 ^b^	82.48 ^ab^	10.28	0.04
Chymotrypsin U/mg	4.73	4.88	4.09	0.85	0.42

AMS = amylase; SEM = standard error of the mean (*n* = 5); CON = a basal diet; 0.1% = a basal diet with 0.1% MSCFA (GML/TB = 1:1); 0.2% = a basal diet with the 0.2% MSCFA (GML/TB = 1:1). ^a,b^ Mean values within a row with different letters differ at *p* < 0.05. And 0.05 ≤ *p* < 0.10 was considered as a tendency.

**Table 8 animals-15-00037-t008:** Effects of MSCFA on intestinal morphology in weaned piglets.

Items	Treatments			SEM	*p*-Value
	CON	0.1%	0.2%		
Duodenum					
Villus height μm	428.91	485.09	436.75	77.30	0.49
Crypt depth μm	486.79	496.02	453.41	82.47	0.70
Villus height/crypt depth	0.88	1.04	1.01	0.22	0.52
Jejunum					
Villus height μm	382.36	442.71	393.24	60.11	0.18
Crypt depth μm	326.20	352.41	392.63	69.35	0.30
Villus height/crypt depth	1.26	1.47	1.15	0.36	0.36
Ileum					
Villus height μm	327.95	368.21	315.26	58.32	0.41
Crypt depth μm	339.75	368.93	365.20	75.48	0.85
Villus height/crypt depth	1.00	1.07	0.91	0.22	0.59

SEM = standard error of the mean (*n* = 5); CON = a basal diet; 0.1% = a basal diet with 0.1% MSCFA (GML/TB = 1:1); 0.2% = a basal diet with the 0.2% MSCFA (GML/TB = 1:1).

## Data Availability

The data presented in this study are available from the corresponding author on request.

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
