# Peer review of "Effects of Medium- and Short-Chain Fatty Acids on Growth Performance, Nutrient Digestibility, Gut Microbiota and Immune Function in Weaned Piglets"

_animals, 2024, doi:10.3390/ani15010037_

Round 1
Reviewer 1 Report
Comments and Suggestions for Authors
The entitled manuscripit "Concentration of α-glycerol monolaurate and tributyrate glyceryl affected growth performance, nutrient digestibility, gut microbiota and immune function of weaned pig" described the using of mediun and short chain fatty acid in pig, and eveluated the effects on growth performance, nutrient digestibility, gut microbiota and immune function. The current version manuscript lack of scientific soundness, and the quality of presentation is with low levels. Here are some advise:
1. The scientific basis of using the combination of α-glycerol and tributyrate glyceryl as nutrition regulation in pigs is required in the introduction section. The current introduction is lack of sufficient.
2. Pigs with both sex was used, are they evenly distributed? Male and female piglets are both in half? these required mentioned in corresponing section.
3. The concentration of MSCFA in the experimental group were 0.1% and 0.2%, what the scientific basis of using these two dose ?
4. The purity of TB is only 60%, the left 40% ingredients content is unknown.
5. Table 1. represent a bad table in nutrition study, please modify with standard layout of each ingredients.
6. The title can be modified to "Effects of MSCFA on growth performance ,,,,,,,,,,in weaned piglets". In addition, all the captions in table or figure shall be modied.
7. Suprisingly, the nutrient apparent digestibility shows significanly improved on day 14 and day 28, why the growth performance had no difference among the groups.
8. Please re-arrenge the serum biochemistry and oxidant status on table 4, so the data presents in a good way.
9. The immunity parameters in the serum has been significant improved by MSCFA, what's the underlying mechanism?
10. The presentation of table 6 must be improved.
11. The tested paramenters is plentiful,but no scienfitic logic among these parameters.
12. The intestinal morphology and microbiota in feces is also very obtrusive. Moreover, the given data did not help explains the mechanism of MSCFA on immunity.
13. The section of discussion is a bit too long. Please shorten in half.
14. in conclusion, the concept of health status is very big. In addition, the author only tested two doses, which can not give a solid conclusion saying that the optimal supplemental concentration is 0.1%.
Author Response
Comments 1: The scientific basis of using the combination of α-glycerol and tributyrate glyceryl as nutrition regulation in pigs is required in the introduction section. The current introduction is lack of sufficient.
Response 1: Thank you for pointing this out. We added relevant content in the introduction section. This change can be found-line: 60-74.
Comments 2: Pigs with both sex was used, are they evenly distributed? Male and female piglets are both in half? these required mentioned in corresponding section.
Response 2: Thanks for your comments. We have revised this as suggested to emphasize this point. This change can be found-line: 94.
Comments 3: The concentration of MSCFA in the experimental group were 0.1% and 0.2%, what the scientific basis of using these two dose ?
Response 3: Thanks for your suggestion. Other studies have found that adding GML or TB alone requires at least 0.1%. We think that the combination may produce better results at lower doses. Therefore, these two doses were initially selected.
Comments 4: The purity of TB is only 60%, the left 40% ingredients content is unknown.
Response 4: Thanks for your comments. We have added the ingredient content about the left 40%. This change can be found-line: 97.
Comments 5: Table 1. represent a bad table in nutrition study, please modify with standard layout of each ingredients.
Response 5: Thanks for your comments. We have revised this as suggested. This change can be found-Table 1.
Comments 6: The title can be modified to "Effects of MSCFA on growth performance ,,,,,,,,,,in weaned piglets". In addition, all the captions in table or figure shall be modied.
Response 6: Thanks for your comments. The title and all the captions in table or figure have been revised this as suggested. This change can be found-Table 1-8.
Comments 7: Surprisingly, the nutrient apparent digestibility shows significantly improved on day 14 and day 28, why the growth performance had no difference among the groups.
Response 7: Thanks for your comments. We have changed the description and made a clear explanation. This change can be found-line: 311-314.
Comments 8: Please re-arrange the serum biochemistry and oxidant status on table 4, so the data presents in a good way.
Response 8: Thanks for your comments. We have revised this as suggested. This change can be found-Table 4.
Comments 9: The immunity parameters in the serum has been significant improved by MSCFA, what's the underlying mechanism?
Response 9: Thanks for your comments. We have changed the description and made a clear explanation. This change can be found-line: 343-346.
Comments 10: The presentation of table 6 must be improved.
Response 10: Thanks for your comments. We have revised this as suggested. This change can be found-Table 6.
Comments 11: The tested paramenters is plentiful,but no scienfitic logic among these parameters.
Response 11: Thanks for your comments. We have changed the description and made a clear explanation. This change can be found among all parameters.
Comments 12: The intestinal morphology and microbiota in feces is also very obtrusive. Moreover, the given data did not help explains the mechanism of MSCFA on immunity.
Response 12: Thanks for your comments. We have changed the description and made a clear explanation.
Comments 13: The section of discussion is a bit too long. Please shorten in half.
Response 13: Thanks for your comments. We have revised this as suggested.
Comments 14: in conclusion, the concept of health status is very big. In addition, the author only tested two doses, which can not give a solid conclusion saying that the optimal supplemental concentration is 0.1%.
Response 14: Thanks for your comments. We have changed the description in conclusion. This change can be found-line: 386-394.

Reviewer 2 Report
Comments and Suggestions for Authors
Dear The authors,
his is a good study; however, more details are required, particularly in the M&M, Results, and Discussion sections. Please refer to the enclosed file for all details.
Best regards,
Reviewer

Author Response
Comments 1: Line 4: “nursery”
Response 1: Thanks for your suggestion. Because this experiment is to study the effects of MSCFA on weaned piglets, I think it should be “weaned”.
Comments 2: Line 14-21:This content comes from the literature review and contains too much information. It should be replaced primarily with the results of the current study.
Line 15:Please replace by ", respectively,".
Line 20-21: It is too broad a conclusion.
Response 2: Thanks for your suggestion. We have revised this manuscript as suggested. This change can be found-line: 23-25.
Comments 3: Line 28:The 0.1% and 0.2% of MSCFA must specify the amounts for both GML and TB.
Response 3: Thanks for your suggestion. We have revised this as suggested. This change can be found-line: 32-33 and 95-96.
Comments 4: Line40: (P > 0.05)
Response 4: Thanks for your suggestion. We have revised this as suggested.
Comments 5: Line 46-47: Please adjust. Please replace "and" by ",".
Response 5: Thanks for your suggestion. We have revised this as suggested. This change can be found-line: 49-50.
Comments 6: Line 55-62:The authors should provide a recommended dose of supplementation.
Response 6: Thanks for your comments. We have revised this as suggested. But parts of them are summary of GML and TB capabilities. Therefore, dosage cannot be added. This change can be found-line: 71.
Comments 7: Line70: Why did the authors choose the 0.1% and 0.2% levels of supplementation? Please provide background information.
Response 7: Thanks for your suggestion. Other studies have found that adding GML or TB alone requires at least 0.1%. We think that the combination may produce better results at lower doses. Therefore, these two doses were initially selected.
Comments 8: Line 79: Please use alphabet format.
Response 8: Thanks for your suggestion. We have revised this as suggested. This change can be found-line: 91.
Comments 9: Line 79: Please provide information on the sex of the piglets studied.
Response 9: Thanks for your suggestion. We have revised this as suggested. This change can be found-line: 94.
Comments 10: Line 82: The full name of MSCFA should be provided somewhere between the introduction section and this point. Please clarify whether GML and TB are classified as short- or medium-chain fatty acids.
Response 10: Thanks for your suggestion. We have revised this as suggested. This change can be found-line: 32 and 96.
Comments 11: Table 1: Metabolizable energy (ME) should be supplied.
Response 11: Thanks for your suggestion. We have revised this as suggested. This change can be found-Table: 1.
Comments 12: Line 100: Please use italic format throughout this manuscript.
Response 12: Thanks for your suggestion. We have revised this as suggested. This change can be found-line: 111.
Comments 13: Line 105: How can it be ensured that all feces from the 8 piglets were collected? Please clarify whether collecting fecal samples for 3 days per period is sufficient.
Response 2: Thanks for your suggestion. We will observe and collect the feces of each piglet by the side of the pen, and the feces mixture collected over three days only needs to be 200g. So, collecting fecal samples for 3 days per period is sufficient.
Comments 14: Line 110: Is this selected piglet included in the digestibility trial?
Response 14: Thanks for your suggestion. We selected piglet included in the digestibility trial.
Comments 15: Line 111: How many quantity?
Response 15: Thanks for your suggestion. The collection of serum is 10 mL. We have revised this as suggested. This change can be found-line: 121.
Comments 16: Line 116: Please specify or indicate the location of each part of the small intestine where samples were collected.
Response 16: Thanks for your suggestion. We have revised this as suggested. This change can be found-line: 126.
Comments 17: Line 129: Is the ADFI calculated based on a per-pen basis? Please provide more information.
Response 17: Thanks for your suggestion. We have revised this as suggested. This change can be found-line: 134-136.
Comments 18: Line 132: The authors should include the results of this analysis instead of the calculations in Table 1.
Response 18: Thanks for your suggestion. We have revised this as suggested. This change can be found- line: 106.
Comments 19: Line 137:Please provide the full name.
Response 19: Thanks for your suggestion. We have revised this as suggested. This change can be found- line: 143.
Comments 20: Line 198: How about nutrient digestibility results?
Response 20: Thanks for your suggestion. We have revised this as suggested. This change can be found- line: 212-213.
Comments 21: Line 203: There are many parameters measured. If possible, p-values above 0.05 should be expressed as 'NS' (not significant).
Response 21: Thanks for your suggestion. We have revised this as suggested. This change can be found- line: 202.
Comments 22: Line 209: How is this calculated? However, based on the statistical analysis results, the authors cannot interpret the results in this way.
Response 22: Thanks for your suggestion. We have deleted this as suggested. This change can be found- line: 202.
Comments 23: Line 221: How about CP digestibility?
Response 23: Thanks for your suggestion. CP digestibility is not significant.
Comments 24: Line 327: Please see the comment in the Results section.
Response 24: Thanks for your suggestion. We have deleted this as suggested. This change can be found- line: 306.
Comments 25: Line 350: In the results, there is no observed influence of MSCFA on intestinal morphology. Please correct this and any related text contents.
Response 25: Thanks for your suggestion. We have revised this as suggested. This change can be found- line: 372-375.
Comments 26: Line 363: Lower scrip for number.
Response 26: Thanks for your suggestion. We have revised this as suggested. This change can be found- line: 338.
Comments 27: Line 372: How is this the case? How did the authors conclude this part?
Response 27: Thanks for your suggestion. We have changed the description and made a clear explanation. This change can be found- line: 339-340.
Comments 28: Line 377: Only IgG? What about other immunoglobulins? In principle, there is a mixture of both passive and active IgGs. Why do the authors believe otherwise?
Response 28: Thanks for your suggestion. We have changed the description and made a clear explanation. This change can be found- line: 346-349.
Comments 29: Line 390-396: Actually, a longitudinal study would provide a better basis for explanation or discussion.
Response 29: Thanks for your suggestion. We have deleted this as suggested. This change can be found- line: 351-352.
Comments 30: Line 414: The authors must focus on explaining the findings based on the results of this study. (Why was there no influence observed in the current study?)"
Response 30: Thanks for your suggestion. We have changed the description and made a clear explanation. This change can be found- line: 372-375.
Comments 31: Line 447: The authors should add a paragraph discussing the integration of all the parameters.
Response 31: Thanks for your suggestion. Your suggestion is very useful in perfecting my manuscript. When we revise the manuscript, the parameters of the discussion have been integrated.
Comments 32: Line 451-454: The conclusion is too broad and does not include the results of the gut microbiota. It must be mentioned in the earlier sections.
Response 32: Thanks for your suggestion. We have changed the description and made a clear explanation. This change can be found- line: 386-394.

Reviewer 3 Report
Comments and Suggestions for Authors
The work by Dong et al. aims to investigate the effect of supplementing 1% and 2% of a mixture of alpha-glycerol monolaurate and tributyrin on zootechnical performance, digestibility of nutritional components, microbiota modulation, and key immunological parameters. The experiments conducted are numerous, and the results are interesting; however, their impact is reduced by the quality of the writing. I recommend an english revision by a native speaker. There are grammatical errors, and the tone is not consistently scientific.
This feedback is intended to be constructive and aims to enhance the study by making it more comprehensible.
Some examples:
Line 60: “growth retardation piglets which leading to enhanced metabolic efficiency in early life” change in thereby leading to…
Line 81: change based with balanced
Table 1: basel = basal
Line 100: “had ad libitum access to feed and water for 28 days. feeders were 100 checked and additional feed a..” improve quality
Line 102: What is meant by "mental state"? Could it refer to behavior? Stereotypies?
Line 205: it is better “ as shown in..” This is simpler and commonly used in scientific writing
Line 219: I don’t understand this sentence” Compared with 0.2% MSCFA, dietary 0.2% MSCFA supplementation increased (P < 0.05) ATTD of DM and OM on d 14 and ATTD of GE on d 28”
Line 384: diminish=reduce
The introduction could be further enhanced to highlight the purpose of the study better.
In Chapter 2.3, "Samples Collection" I don't understand why the samples were first dried and then allowed to reabsorb moisture before being ground. What is the scientific rationale behind this? At most, I could understand them reabsorbing some ambient moisture, but I don't understand the meaning behind this step.
The discussion is well-structured, but difficult to read. It is recommended that clarity is improved by adopting a more academic writing style.
Comments on the Quality of English Language
It is recommended to revise the English to improve the academic style and clarity.
Author Response
Comments 1: Line 60: “growth retardation piglets which leading to enhanced metabolic efficiency in early life” change in thereby leading to…
Response 1: Thanks for your suggestion. We have revised this as suggested. This change can be found-line: 61.
Comments 2: Line 81: change based with balanced
Response 2: Thanks for your suggestion. We have revised this as suggested. This change can be found-line: 92- 94.
Comments 3: Table 1: basel = basal
Response 3: Thanks for your suggestion. We have revised this as suggested. This change can be found-Table 1.
Comments 4: Line 100: “had ad libitum access to feed and water for 28 days. feeders were 100 checked and additional feed a..” improve quality
Response 4: Thanks for your suggestion. We have revised this as suggested. This change can be found-line: 110-111.
Comments 5: Line 102: What is meant by "mental state"? Could it refer to behavior? Stereotypies?
Response 5: Thanks for your suggestion. It should be health state. We have revised as suggested. This change can be found-line: 113.
Comments 6: Line 205: it is better “ as shown in..” This is simpler and commonly used in scientific writing
Response 6: Thanks for your suggestion. We have revised this manuscript as suggested.
Comments 7: Line 219: I don’t understand this sentence” Compared with 0.2% MSCFA, dietary 0.2% MSCFA supplementation increased (P < 0.05) ATTD of DM and OM on d 14 and ATTD of GE on d 28”
Response 7: Thanks for your comments. We have revised this as suggested. This change can be found-line: 212-213.
Comments 8: Line 384: diminish=reduce
Response 8: Thanks for your comments. We have revised this as suggested.
Comments 9: The introduction could be further enhanced to highlight the purpose of the study better.
Response 9: Thanks for your comments. We have changed the description and made a clear explanation. This change can be found-line: 81-84.
Comments 10: In Chapter 2.3, "Samples Collection" I don't understand why the samples were first dried and then allowed to reabsorb moisture before being ground. What is the scientific rationale behind this? At most, I could understand them reabsorbing some ambient moisture, but I don't understand the meaning behind this step.
Response 10: Thanks for your comments. Samples were first dried and then allowed to reabsorb moisture before being ground. This is the standard procedure for drying samples. When we measure moisture, we can avoid the influence of moisture in the air. The measurement data is more accurate.
Comments 11: The discussion is well-structured, but difficult to read. It is recommended that clarity is improved by adopting a more academic writing style.
Response 11: Thanks for your comments. We have changed the description and made a clear explanation by adopting a more academic writing style.

Round 2
Reviewer 3 Report
Comments and Suggestions for Authors
Thank you for your answers.
The manuscript has been correctly revised.